# TPI-LLM: Serving 70B-scale LLMs Efficiently on Low-resource Edge Devices

## Abstract

Large model inference is shifting from cloud to edge due to concerns about the privacy of user interaction data. However, edge devices often struggle with limited computing power, memory, and bandwidth, requiring collaboration across multiple devices to run and speed up LLM inference. Pipeline parallelism, the mainstream solution, is inefficient for single-user scenarios, while tensor parallelism struggles with frequent communications. In this paper, we argue that tensor parallelism can be more effective than pipeline on low-resource devices, and present a compute- and memory-efficient tensor parallel inference system, named TPI-LLM, to serve 70B-scale models. TPI-LLM keeps sensitive raw data local in the users' devices and introduces a sliding window memory scheduler to dynamically manage layer weights during inference, with disk I/O latency overlapped with the computation and communication. This allows larger models to run smoothly on memory-limited devices. We analyze the communication bottleneck and find that link latency, not bandwidth, emerges as the main issue, so a star-based allreduce algorithm is implemented. Through extensive experiments on both emulated and real testbeds, TPI-LLM demonstrated over 80% less time-to-first-token and token latency compared to Accelerate, and over 90% compared to Transformers and Galaxy, while cutting the peak memory footprint of Llama 2-70B by 90%, requiring only 3.1 GB of memory for 70B-scale models.

## 1 Introduction

Recently, Large Language Models (LLMs) have been widely deployed in the cloud for inference. User inputs are uploaded to the cloud, where high-performance GPUs are used to compute output sequences, and then sent back to user devices for display. This process poses privacy risks, as user prompts are exposed to network intermediaries and clouds. Therefore, there is an increasing need to shift LLM services to the network edge, such as on laptops, hand phones, tablets, and desktop computers. However, edge devices have very limited memory (4-16 GB) and computing power (often CPU-only). Even with quantization, running a Llama 3.1-70B model still requires at least 40 GB of memory, which far exceeds the capacity of most edge devices. Besides, running Bert-L on one Nano-M device results in a latency that is $120\times$ longer than on one A100 GPU. This gap requires the use of more edge devices to support and speed up LLM inference on the network edge.

While advanced LLM serving systems (Shoeybi et al., 2019; Rasley et al., 2020; Li et al., 2023; Agrawal et al., 2024; Miao et al., 2024) have been designed for high-performance GPU clusters, recent efforts (Zhang et al., 2024; Mei et al., 2024; Borzunov et al., 2024) are adapting these systems to edge environments, by adaptively partitioning model between edge devices and optimizing schedulers to boost token throughput. However, in smart home scenarios like smart speaker, edge LLM systems often handle one user request at a time, making them degrade from pipeline to model parallelism and leaving devices idle most of the time. Thus, tensor parallelism is preferred for better efficiency. For instance, Ye et al. (2024) combine tensor and sequence parallelism to reduce token latency and Wei et al. (2024) introduce block parallelism to restructure Transformer layers.

However, even with 8 devices sharing the load, running full-precision Llama 2-70B still requires 35 GB per device, memory remains a shortage. Solutions like memory block paging (Kwon et al., 2023) and optimized KVCache storage (Jin et al., 2023; Lee et al., 2024) help schedule data between GPUs and CPUs, but unfortunately, GPUs are not available on most edge devices. As a popular alternative,

Accelerate (Gugger et al., 2022) can offload model data from a CPU to a disk to run larger models, but its blocking I/O drastically slows inference, with token latency increases to 30 seconds per token on Llama 3.1-8B.

In this work, we analyze why tensor parallelism is more effective than model parallelism on low-resource edge devices and present TPI-LLM, a computing- and memory-efficient tensor parallel inference framework to serve LLM models. Constrained by the high link latency, a star-based allreduce algorithm is implemented. To address the memory shortage, a sliding window memory scheduler is further introduced. We build a prototype of TPI-LLM with 3K LoC and two testbeds using Klonet (Ma et al., 2024) and 4 laptops. Extensive results on Llama 3.1-8B/70B (Dubey et al., 2024), Llama 2-3B/7B/13B/70B (Touvron et al., 2023) and Yi-34B (AI et al., 2024) demonstrate the significant reduction of the memory footprint and faster inference speed compared to Transformers (Wolf et al., 2020), Acclerate (Gugger et al., 2022), and Galaxy (Ye et al., 2024).

We summarize the main contributions of this work as follows:

- We design a TPI-LLM for edge LLM serving, which keeps prompt privacy in mind to allow edge devices with limited computing power collaborate to deliver faster inference.
- We find that network bandwidth is no longer an issue. Instead, link latency causes high delays in advanced allreduce algorithms. Thus, a star-based allreduce algorithm is implemented, which greatly outperforms ring- and tree-based methods.
- We introduce a sliding window memory scheduler, which asynchronously loads and unloads layer weights and overlaps disk I/O latency with computations and communications, enabling the inference of larger models on low-memory devices.
- We prototype TPI-LLM and show that it reduces time-to-first-token and token latency by over 80% compared to Accelerate and over 90% compared to Transformers and Galaxy. It serves Llama 2-70B with a peak memory footprint of 3.1 GB across 8 low-resource devices.

## 2 OBSERVATIONS AND MOTIVATIONS

Before presenting our TPI-LLM system, we address two questions that guide our design:

*Q1: On low-resource edge devices, which dominate inference time: computation or communication? Which is more efficient, tensor parallelism or model parallelism?*

On the network edge, the balance between computation and communication differs from that in high-performance GPU clusters. To determine whether tensor or model parallelism offers more benefits, it is essential to identify which—computation or communication—takes up more time. For this purpose, we examine the Llama 3.1-8B model on a LAN network with 4 laptops of 8 cores. The network bandwidth between them is 178 Mbps, and the devices implement allreduce communications using a parameter server architecture (Li et al., 2014).

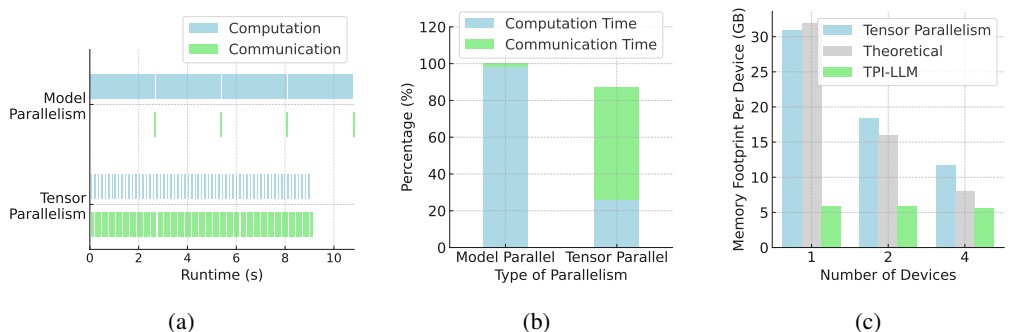

Figure 1: Comparison of (a,b) tensor and model parallelism in terms of computational and communication time and (c) memory footprint each device with increasing tensor parallel nodes.

Figures 1a and 1b show the timeline and computing-communication time ratio for model and tensor parallelism during inference. In model parallelism, communication accounts for only 2% of the

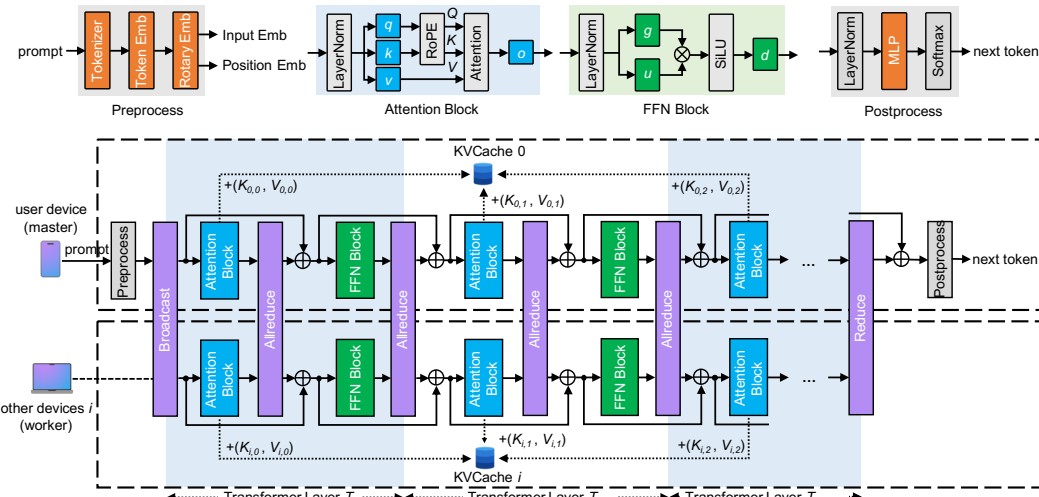

Figure 2: Overview of the TPI-LLM parallel framework.

time, with most spent on computation. However, when one device is computing, others are idle, creating pipeline bubbles and resource waste. In tensor parallelism, communication rises to 70%, but all devices compute simultaneously, and the speed boost outweighs the communication cost, leading to less overall inference time. This makes tensor parallelism the preferred choice.

*Q2: Is tensor parallelism enough for edge LLM serving?*

Tensor parallelism does reduce memory footprint each device by sharing model parameters across multiple devices, but it doesn't fully address the memory shortage. Figure 1c shows that even with 4 tensor parallel nodes, memory footprint remains at 12 GB—still too high for most edge devices. This is because memory footprint includes not just model parameters but also intermediate results, key value cache, libraries, etc., causing the actual usage to exceed the theoretical value. Besides, other apps on the device also compete for memory, which worsens the shortage. Thus, even with tensor parallelism, a memory scheduler is still needed to avoid out-of-memory (OOM) issues.

## 3 TPI-LLM FRAMEWORK WITH SLIDING WINDOW MEMORY SCHEDULING

In a typical inference workflow, many users send their prompts to a cloud-based service. These prompts are pooled and scheduled in batches, undergoing dozens of Transformer layers, and converted into probabilities to predict the next token. This process repeats until the generated sequence is finished. While the fundamental workflow on the cloud and edge are similar, key differences arise:

*(a) Keep prompts and generated sequences on users' device.* In a cloud setup, user prompts are sent to remote servers for processing, which result in exposure of private data. Edge LLM serving systems are required to keep prompts and generated sequences in users' own devices to ensure raw data never get exposed to external unknown environments.

*(b) More single-prompt serving.* Current LLM serving systems are typically optimized for batched prompts using pipeline scheduling. However, these optimizations lead to resource underutilization in edge scenarios like smart speakers, where only one prompt is processed at a time.

*(c) Low-resource devices without CUDA support.* Edge devices, unlike cloud GPUs, have very limited memory and low computing power. Many of them lack CUDA support or do not have GPUs at all, and they often prioritize full precision to ensure faster computations.

### 3.1 THE PARALLEL FRAMEWORK DESIGN OF TPI-LLM SYSTEM

The proposed tensor parallel inference system (TPI-LLM) tackles these challenges by using a tensor parallel framework that distributes attention heads across multiple nodes. As depicted in Figure 2,

**Algorithm 1:** Master (with rank 0):

1 Split and distribute pretrained weight files to worker nodes;
2 Tokenize user prompt into indices;
3 Start memory scheduler;
4 **while** *generated sequence not finished* **do**
5    *Preprocess:* Convert indices to input and position embeddings, calculate causal mask and cache position;
6    *Broadcast:* Send embeddings, causal mask, and cache position to workers;
7    **foreach** *decoder layer l* **do**
8       *Attention:* Execute layernorm, self-attention, and store $(K_l^0, V_l^0)$ in KVCache $\mathcal{D}^0$;
9       *Allreduce:* Aggregate attention outputs;
10      *FFN:* Execute layernorm and FFN;
11      *Allreduce:* Aggregate FFN outputs;
12    **end**
13    *Reduce:* Sum final outputs with others;
14    *Postprocess:* Execute layernorm, MLP, softmax, and sample next token;
15 **end**

**Algorithm 2:** Worker (with rank $k$):

1 Download sliced weight files from the master node;
2 Start memory scheduler;
3 **while** *generated sequence not finished* **do**
4    *Broadcast:* Receive embeddings, causal mask, and cache position from master;
5    **foreach** *decoder layer l* **do**
6       *Attention:* Execute layernorm, self-attention, and store $(K_l^k, V_l^k)$ in KVCache $\mathcal{D}^k$;
7       *Allreduce:* Aggregate attention outputs;
8       *FFN:* Execute layernorm and FFN;
9       *Allreduce:* Aggregate FFN outputs;
10    **end**
11    *Reduce:* Send final output to master;
12 **end**

it involves a master node, typically the user's device that initiates the prompt, and several worker nodes that share the computational load. Their pseudo codes are given in Algorithms 1 and 2.

*Step 1: The master node partitions and distributes model weights.* Before inference begins, the master node partitions the pretrained model weights $\boldsymbol{W}$, such as attention heads and FFN weights, among the worker nodes. Workers with greater computing power and larger memory are allocated more attention heads and FFN weights. This ensures no single device bears the full burden.

*Step 2: The master node initiates prompt and broadcast the input embedding to workers.* The inference process starts at the master node, where a user prompt is tokenized into a list of token indices $\boldsymbol{x}$ and then transformed into input embeddings $\boldsymbol{H}^0 = \boldsymbol{x}\boldsymbol{W}_{\text{emb}}$. The embedding is then broadcast to all worker nodes $\boldsymbol{H}_{\text{ffn}}^0 = \boldsymbol{H}^0$ to initiate the tensor parallel workflow.

*Step 3: All nodes perform tensor parallel computing.* The tensor parallel computing follows a cycle of four operations: attention computing, allreduce, FFN computing, and allreduce. These operations together constitute a Transformer block. Devices compute attention and FFN with partitioned weights in parallel, reducing the computing delays on low-power devices.

In the attention computation phase of the $l$-th Transformer block, device $h$ processes only a subset of attention heads $\boldsymbol{Q}^{h,l} = \boldsymbol{H}_{\text{norm}}^l \boldsymbol{W}_Q^{h,l}$, $\boldsymbol{K}^{h,l} = \boldsymbol{H}_{\text{norm}}^l \boldsymbol{W}_K^{h,l}$, $\boldsymbol{V}^{h,l} = \boldsymbol{H}_{\text{norm}}^l \boldsymbol{W}_V^{h,l}$, where $\boldsymbol{H}_{\text{norm}}^l = \text{norm}(\boldsymbol{H}_{\text{ffn}}^{l-1})$ is the normed hidden state and weight partitions $\boldsymbol{W}_Q^{h,l}, \boldsymbol{W}_K^{h,l}, \boldsymbol{W}_V^{h,l}$ are downloaded from the master node in Step 1. Once $\boldsymbol{Q}^{h,l}, \boldsymbol{K}^{h,l}, \boldsymbol{V}^{h,l}$ are computed, we apply the scaled dot-product attention to calculate the attention score, and the result is then synchronized across devices:

$$\boldsymbol{H}_{\text{attn}}^l = \texttt{all\_reduce}(\texttt{softmax}(\frac{\boldsymbol{Q}^{h,l}(\boldsymbol{K}^{h,l})^\top}{\sqrt{d}})\boldsymbol{V}^{h,l}) + \boldsymbol{H}_{\text{ffn}}^{l-1}, \tag{1}$$

where $d$ is the dimension for attention head. Here, attention is computed in parallel across devices, followed by an allreduce to aggregate their hidden states and a shortcut connection. The key-value pair $(\boldsymbol{K}^{h,l}, \boldsymbol{V}^{h,l})$ is cached locally on device $h$ to reduce redundant computations. This distributed KVCache partitions the cache across devices, so memory cost is reduced on individual device.

After the attention computation and allreduce, the process continues with the FFN computation:

$$\boldsymbol{H}_{\text{ffn}}^l = \texttt{all\_reduce}(\boldsymbol{W}_{\text{down}}^{h,l} \cdot (\sigma(\boldsymbol{W}_{\text{gate}}^{h,l} \cdot \boldsymbol{H}_{\text{norm}}^l) \odot (\boldsymbol{W}_{\text{up}}^{h,l} \cdot \boldsymbol{H}_{\text{norm}}^l))) + \boldsymbol{H}_{\text{attn}}^l, \tag{2}$$

where FFN weights $\boldsymbol{W}_{\texttt{gate}}^{h,l}, \boldsymbol{W}_{\texttt{up}}^{h,l}, \boldsymbol{W}_{\texttt{down}}^{h,l}$ are also partitioned weights, $\boldsymbol{H}_{\texttt{norm}}^l = \texttt{norm}(\boldsymbol{H}_{\texttt{attn}}^l)$, $\sigma$ represents the activation function such as SiLU (Elfwing et al., 2018). Similar to the attention computation stage, the FFN is computed in parallel, followed by an allreduce and a shortcut connection.

*Step 4: The master node reduces tensor parallel results and calculates the next token.* After each node $h$ completes its part of computation within the backbone network, the result is sent to the master node. The summed results $\boldsymbol{H}_{\texttt{ffn}}^L$ are then passed through a task head $\boldsymbol{W}_{\texttt{head}}$ and softmax to obtain the probability distribution of the next token $\boldsymbol{z} = \texttt{softmax}(\boldsymbol{H}_{\texttt{ffn}}^L \boldsymbol{W}_{\texttt{head}})$, which is then sampled. Steps 2 to 4 repeat until an EOS token is generated or the length limit is reached.

TPI-LLM provides three benefits: (i) The user prompt $\{\boldsymbol{x}_1, \boldsymbol{x}_2, \cdots\}$ and generated sequence $\{z_1 \sim \boldsymbol{z}_1, z_2 \sim \boldsymbol{z}_2, \cdots\}$ are processed only on the master node, keeping them hidden from workers. Even if workers reverse-engineer input embeddings $\boldsymbol{H}^0$, they cannot recover the raw prompt $\boldsymbol{x}$ or next token $z \sim \boldsymbol{z}$ since the weights of input embedding $\boldsymbol{W}_{\texttt{emb}}$ and task head $\boldsymbol{W}_{\texttt{head}}$ reside solely on master. (ii) The inference speed is often limited by the computational latency, but in TPI-LLM, it is accelerated via parallel computing. (iii) Unlike other systems that use a mix of communication primitives (reduce & broadcast (Shoeybi et al., 2019), reducescatter & allgather (Ye et al., 2024), etc.), TPI-LLM standardizes communications to allreduce. This enhances compatibility with broader communication libraries like PS-LITE (Chen et al., 2015) and NetStorm (Li et al., 2024), leveraging their optimized implementations for edge conditions.

## 3.2 ALLREDUCE LATENCY ANALYSIS

Given the dynamic and heterogeneous nature of edge networks, we tested NetStorm (Li et al., 2024) as the communication backend, but unfortunately, it resulted in high token latency. After further validation, we confirmed that this latency was not due to network bandwidth, but due to link latency.

To analyze the impact of network bandwidth and link latency, we make the following assumption.

**Assumption 1.** *Assume that the edge network adopts a physical topology as shown in Appendix A.7, the network links have the same latency $\tau$, the allreduce algorithm follows a tree-based structure of depth 2 for aggregation, and each device has the same computing power.*

The allreduce latency can be expressed as $t_{\texttt{all\_reduce}} = 2L(t_{\texttt{data}} + t_{\texttt{link}} + t_{\texttt{barrier}} + t_{\texttt{aggr}})$, where $L$ is the number of Transformer layers, $t_{\texttt{data}}$ is the cumulative data transfer latency, $t_{\texttt{link}}$ is the cumulative link latency, $t_{\texttt{barrier}}$ is the cumulative barrier latency during aggregation, and $t_{\texttt{aggr}}$ is the cumulative latency for aggregation calculation. Here we ignore $t_{\texttt{aggr}}$ as it takes only 0.1 ms and thus negligible compared to other factors.

**Proposition 1.** *The bottleneck in allreduce is not network bandwidth, but link latency.*

*Proof.* The data transfer latency $t_{\texttt{data}} = 2\sum_{\{i \to j\} \in \mathcal{P}_h} \frac{32|\boldsymbol{H}|}{B_{ij}}$ depends on the size $32|\boldsymbol{H}|$ of the data being transmitted and the bandwidth $B_{ij}$ of the links in the path $\mathcal{P}_h$, here $\mathcal{P}_h$ is an index sequence from device $h$ to the master device. For example, in the case of Llama 2-70B with a hidden size $|\boldsymbol{H}| = 8192$ and a network bandwidth of 300 Mbps, the data transfer latency is only $t_{\texttt{data}} = 3.4$ ms, which is negligible compared to other latencies. In addition, experiment results in Figure 5 show that increasing the network bandwidth does not significantly reduce token latency, further confirming that data transfer and network bandwidth is not the bottleneck.

The link latency $t_{\texttt{link}}$, which is often neglected, emerges as the main issue. For example, the path from device $h_2$ to $h_1$ via $h_8$ follows the route $h_2 \to r_2 \to r_9 \to r_8 \to h_8 \to r_8 \to r_9 \to r_1 \to h_1$, resulting in a total link latency of $16\tau$, where $\tau$ is the per-hop link latency. To isolate the impact of link latency, we ran allreduce with only 4 bytes of data, excluding data transfer $t_{\texttt{data}}$ and barrier latencies $t_{\texttt{barrier}}$. The results, shown in Figure 3, demonstrate that the per-link latency $\tau$ significantly impacts the allreduce latency. This indicates that an inefficient allreduce algorithm, where multiple hops are required (e.g., ring (Ye et al., 2024; Shoeybi et al., 2019) or tree-based (Zhou et al., 2021; Li et al., 2024) algorithms), will further amplifies this impact. For example, with the ring algorithm, allreduce requires

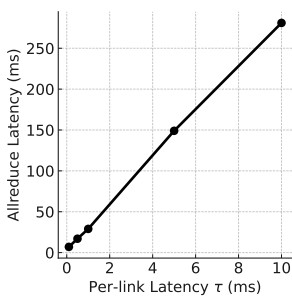

Figure 3: Impact of link latency $\tau$.

7 communication steps for reducescatter and 7 for allgather, resulting in a total link latency of $56\tau$, which is $3.5\times$ higher than the tree-based setup.

The barrier latency, $t_{\text{barrier}}$, arises from synchronization delays during data aggregation. Given the assumption that all devices have equal computing power and network links have equal latencies, the barrier latency can be approximated as negligible:

$$t_{\text{barrier}} = \max\{\sum_{(i \to j) \in \mathcal{P}} \frac{32|\boldsymbol{H}|}{B_{ij}}, \forall \mathcal{P}\} - \min\{\sum_{(i \to j) \in \mathcal{P}} \frac{32|\boldsymbol{H}|}{B_{ij}}, \forall \mathcal{P}\} \approx 0. \tag{3}$$

Thus, link latency $t_{\text{link}}$ emerges as the key factor in allreduce latency. $\qquad\square$

**Proposition 2.** *The star-based allreduce is more effective for TPI-LLM in high-latency networks.*

Despite past criticism, the star-based allreduce, where workers push data directly to the master for aggregation and pull the result back (Chen et al., 2015), stands out as the best choice (see Appendix A.1 for a detailed proof). It has minimal hops (8), lowest link latency ($8\tau$), zero intermediate barriers, and avoids the single-point issue due to the small data size (256 KB per device), making it the preferred allreduce algorithm for TPI-LLM.

### 3.3 Sliding Window Memory Scheduling

Quantizations like FP16 and INT8 are common for NVIDIA GPUs with CUDA support, but most edge devices lack CUDA and prefer full precision for faster computation due to their general-purpose CPU design. As a result, while tensor parallelism helps distribute memory costs across devices, the memory load remains high. Thus, memory scheduling is still required to manage these loads.

We introduce a memory scheduler, which manages memory by dynamically loading and unloading model weights during inference, ensuring that only the necessary parts are kept in memory (see Appendix A.2 for potential use). The memory scheduler operates on a daemon thread to asynchronously handle memory operations. To maintain the peak memory footprint, it uses a sliding window and preloads weights for upcoming layers while unloading those that have been processed.

As mentioned in Section 3.1, each Transformer layer is divided into attention computing, allreduce, FFN computing, and allreduce. For simplicity, in Figure 4, we assume the delays for these stages and weight loading to be equal. In each time slot, the memory scheduler asynchronously loads weights for either an attention or FFN block. By overlapping weight loading with ongoing computations and communications, it hides the I/O latency associated with loading weights from disk. For example, in Figure 4, the memory scheduler loads one more block during each allreduce until the sliding window reaches its size. As computations and communications proceed, we ensure weights are always ready when needed, allowing for seamless inference without computational stalls.

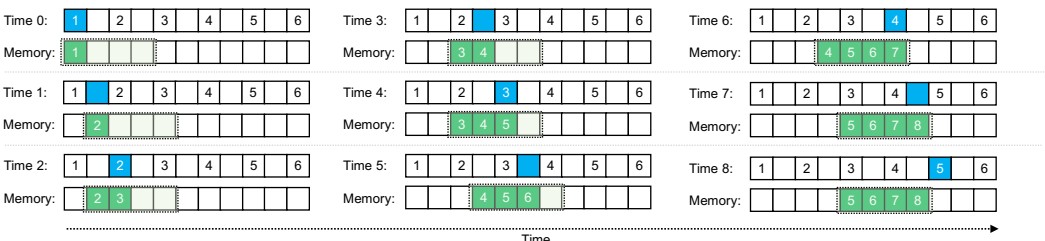

Figure 4: An illustration of the sliding window memory scheduling. Blue blocks indicate the blocks currently executed, with numbered blocks for attention or FFN computing and unnumbered blocks for allreduce communication. Green blocks indicate loaded model weights. The dashed box represents the sliding window, with size 4 in this case.

Next, we provide the conditions for this mechanism to reach a steady state, under which all required weights are loaded before computation starts.

**Proposition 3** (**Loose Steady Condition**). *The memory scheduler reaches a steady state when the following condition is met:*

$$t_{\text{attn}} + t_{\text{ffn}} + 2t_{\text{all\_reduce}} \geq \tau_{\text{ffn}} + \tau_{\text{attn}}, \tag{4}$$

*and one of the following conditions is met:*

$$l \cdot t_{\texttt{attn}} + (l-1) \cdot t_{\texttt{ffn}} + (2l-1) \cdot t_{\texttt{all\_reduce}} \geq l \cdot \tau_{\texttt{ffn}} + (l-1) \cdot \tau_{\texttt{attn}}, \ \forall l \in \{1, \cdots, L\}, \quad (5)$$

$$(l-1) \cdot t_{\texttt{attn}} + l \cdot t_{\texttt{ffn}} + (2l-1) \cdot t_{\texttt{all\_reduce}} \geq (l-1) \cdot \tau_{\texttt{ffn}} + l \cdot \tau_{\texttt{attn}}, \ \forall l \in \{1, \cdots, L\}, \quad (6)$$

*where $t_{\texttt{attn}}$ and $t_{\texttt{ffn}}$ are times required for attention and FFN computation, $t_{\texttt{all\_reduce}}$ is the allreduce latency, $\tau_{\texttt{ffn}}$ and $\tau_{\texttt{attn}}$ are times required to load attention and FFN weights, and $L$ is the number of Transformer layers.*

This condition is loose but a bit hard to assess, so we present a tighter, more intuitive condition.

**Proposition 4 (Tight Steady Condition).** $t_{\texttt{attn}} + t_{\texttt{all\_reduce}} \geq \tau_{\texttt{ffn}}$ *and* $t_{\texttt{ffn}} + t_{\texttt{all\_reduce}} \geq \tau_{\texttt{attn}}$.

The proofs can be found in Appendices A.3 and A.4. This conclusion is straightforward. If the previous block's computation and allreduce time cover the current block's weight loading time, the memory scheduler can fully hide the disk I/O latency. As an example, in Section 4.4, we use 4 laptops with Llama 2-7B, setting $p_i = 0.25$ and $w = 4$. We measured $t_{\texttt{attn}} = 11$ ms, $t_{\texttt{ffn}} = 17$ ms, $t_{\texttt{all\_reduce}} = 14$ ms, $\tau_{\texttt{attn}} = 18$ ms, and $\tau_{\texttt{ffn}} = 30$ ms. While the tight steady condition is not met, the loose steady condition is met, allowing the memory scheduler to achieve steady state.

**Proposition 5 (Peak Memory Footprint).** *If the memory scheduler reaches a steady state, the peak memory footprint of the master and worker can be expressed as*

$$M_{\texttt{master}} = \gamma \times \begin{cases} hv + h, & \text{if } w = 1 \\ 2hv + h, & \text{if } w = 2 \\ 2hv + h + \lfloor \frac{w-2}{2} \rfloor \left( 2(1 + \frac{b}{a})h^2 p_i + h \right) + \lfloor \frac{w-1}{2} \rfloor (3hsp_i + h), & \text{if } w \geq 3 \end{cases} \quad (7)$$

$$M_{\texttt{worker}} = \gamma \times \left( \lfloor \frac{w}{2} \rfloor \left( 2(1 + \frac{b}{a})h^2 p_i + h \right) + \lfloor \frac{w+1}{2} \rfloor (3hsp_i + h) \right), \quad (8)$$

*where $h$ is the hidden size, $v$ is the vocabulary size, $a$ is the number of attention heads, $b$ is the number of key-value heads, $s$ is the intermediate size, $p_i$ is the proportion of parameters handled by device $i$, $w$ is the memory window size, and $\gamma$ is a memory scaling factor.*

The proof can be found in Appendix A.5. However, if a slow disk I/O disrupts the steady state, the memory scheduler will retain some FFN blocks in memory to reduce disk access frequency.

**Proposition 6 (Loose Steady Condition with Block Retention).** *Let the memory scheduler retain one FFN block in memory every $T$ FFN blocks, the condition to reach a steady state is then*

$$l \cdot t_{\texttt{attn}} + l \cdot t_{\texttt{ffn}} + 2l \cdot t_{\texttt{all\_reduce}} \geq (l - \lceil \frac{l}{T} \rceil) \cdot \tau_{\texttt{ffn}} + l \cdot \tau_{\texttt{attn}}, \quad (9)$$

$$l \cdot t_{\texttt{attn}} + (l-1) \cdot t_{\texttt{ffn}} + (2l-1) \cdot t_{\texttt{all\_reduce}} \geq (l - \lceil \frac{l}{T} \rceil) \cdot \tau_{\texttt{ffn}} + (l-1) \cdot \tau_{\texttt{attn}}. \quad (10)$$

The proof can be found in Appendix A.6. By setting an appropriate $T$, idle memory can help the scheduler reach a steady state, thus achieving a tradeoff between memory use and inference speed.

## 4 EXPERIMENTS

*Prototype and Testbed.* We implemented the prototype of TPI-LLM[1] with 3K LoC using PyTorch and Transformers to provide flexible support for various sizes and versions of pretrained LLMs. Our testbed, illustrated in Appendix A.7, was built upon Klonet (Ma et al., 2024) to create an edge network environment, emulating realistic conditions with configurable properties like network topology, bandwidth, and latency. By default, 8 edge devices were emulated on 2 Intel Xeon Gold 5220R CPUs, each limited to 8 logical cores, 8 GB of memory, and 4 GB of swap. Network bandwidth between devices was set to 300 Mbps with a 1 ms latency.

*Models.* The inference speed of TPI-LLM is significantly affected by the model architecture. Deeper layers, more parameters, larger hidden sizes, and more attention heads increase the computational latency. Additionally, deeper layers result in more allreduce communications, and a larger hidden size leads to greater traffic. We tested with various models of different sizes, including Llama 2-3B/7B/13B/70B, Llama 3-8B/70B, and Yi-34B. See Appendix A.8 for their configuration details.

---

[1]Open available at: `https://anonymous.4open.science/r/tpi-llm`.

Table 1: The TTFT, token latency, and peak memory footprint per device of TPI-LLM.

| Model (FP32) | Memory Scheduler Disabled | | | Memory Scheduler Enabled | | |
|---|---|---|---|---|---|---|
| | TTFT | Latency | Memory | TTFT | Latency | Memory |
| Llama 2-3B | 2.3 s | 1.0 s/token | 2.8 GB | 2.0 s | 1.9 s/token | 1.4 GB |
| Llama 2-7B | 3.1 s | 1.2 s/token | 4.5 GB | 3.0 s | 2.6 s/token | 1.7 GB |
| Llama 2-13B | 5.1 s | 1.9 s/token | 8.1 GB | 5.8 s | 2.9 s/token | 2.1 GB |
| Llama 2-70B | OOM | OOM | 34.9 GB | 29.4 s | 26.1 s/token | 3.1 GB |
| Llama 3.1-8B | 4.5 s | 1.5 s/token | 8.5 GB | 4.5 s | 4.3 s/token | 5.4 GB |
| Llama 3.1-70B | OOM | OOM | 42.3 GB | 32.9 s | 29.9 s/token | 11.3 GB |
| Yi-34B | OOM | OOM | 20.4 GB | 15.7 s | 13.7 s/token | 4.9 GB |

Table 2: Peak memory footprint per device with the memory window size set to 2.

| Model (FP32) | Memory Scheduler Disabled (GB) | | | | Memory Scheduler Enabled (GB) | | | |
|---|---|---|---|---|---|---|---|---|
| | $N=2$ | $N=4$ | $N=6$ | $N=8$ | $N=2$ | $N=4$ | $N=6$ | $N=8$ |
| Llama 2-3B | 7.3 | 4.3 | 3.2 | 2.8 | 1.5 | 1.4 | 1.4 | 1.4 |
| Llama 2-7B | 13.7 | 7.7 | 5.5 | 4.5 | 2.0 | 1.8 | 1.7 | 1.7 |
| Llama 2-13B | 25.7 | 13.9 | 9.8 | 8.1 | 2.3 | 2.2 | 2.2 | 2.1 |
| Llama 2-70B | 130 | 66.6 | 46.6 | 34.9 | 3.7 | 3.3 | 3.3 | 3.1 |
| Llama 3.1-8B | 18.4 | 11.8 | 9.4 | 8.5 | 5.9 | 5.6 | 5.5 | 5.4 |
| Llama 3.1-70B | 137.7 | 74.0 | 51.1 | 42.3 | 10.8 | 10.5 | 11.4 | 11.3 |
| Yi-34B | 67 | 36.4 | 23.9 | 20.4 | 5.0 | 5.0 | 5.0 | 4.9 |

## 4.1 OVERVIEW OF TPI-LLM PERFORMANCE

**Fit 70B LLMs into edge devices and run in high efficiency.** We tested the performance of TPI-LLM with a focus on 3 key metrics: time-to-first-token (TTFT), token latency, and peak memory footprint per device. The memory window size is set to 2 by default. As shown in Table 1, without the memory scheduler, the full weights are loaded into the memory at once. Despite that these weights have been distributed across multiple devices, the memory is still insufficient for larger models like Yi-34B and Llama 2/3/3.1-70B. Instead, enabling our memory scheduler significantly reduces the peak memory footprint, allowing larger models to run efficiently. For example, the Llama 2-70B model requires just 3.1 GB of memory per device, and the Llama 3.1-70B model fits within device limits. The results are summarized in Table 1.

**No need for dozens of devices, one or two are enough to run 70B models.** We used 8 devices by default, but can fewer devices run 70B-scale models? Table 2 gives detailed peak memory footprints with varying number of devices. Without the memory scheduler, full weights are loaded onto the devices, and with fewer devices, the memory load increases. For instance, using only 2 devices limits users to smaller models, like those between 3B and 7B. However, with the memory scheduler enabled, only a few layers' weights are loaded and distributed across devices. This allows even larger models, such as 70B, to run smoothly on just 2 devices. Appendix A.9 shows the case with a memory window size of 4, which requires slightly more memory but faster speed. The peak memory footprint in TPI-LLM is primarily determined by the product of vocabulary size and hidden size, which is detailed in equation (7) and can be further reduced in our future work.

## 4.2 SCALING OVER VARYING EDGE CONDITIONS

**Computation remains the bottleneck, not network bandwidth.** In this experiment, we examined the token latency of TPI-LLM under different edge conditions, the results are shown in Figure 5. As expected, increasing the number of devices reduces the computing load on each device, significantly lowering token latency, and more CPU cores also contribute to a reduced latency. Instead, a limited network bandwidth was no longer a bottleneck, boosting it from 300 Mbps to 1 Gbps had little effect on latency due to the tiny data size (only 256 KB) during each allreduce. Thus, the main bottleneck remains in the computation, which our future work should focus on.

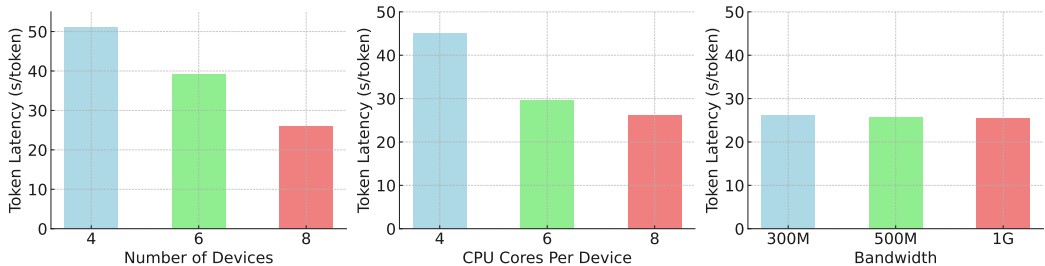

Figure 5: Token latency over varying number of devices, CPU cores, and network bandwidth on Llama 2-70B.

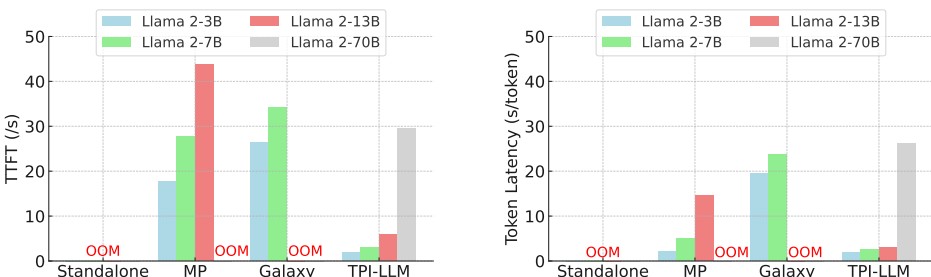

Figure 6: Comparison of TPI-LLM with three benchmarks.

## 4.3 COMPARISON WITH BENCHMARKS

We compared the TPI-LLM with 3 benchmarks: (a) *Standalone:* LLM inference is executed only on a single edge device using Transformers (Wolf et al., 2020). (b) *Model Parallelism (MP):* Since only one user is served at a time, the pipeline parallelism (Zhang et al., 2024; Mei et al., 2024; Borzunov et al., 2024) degrades to the model parallelism, where different layer sequences are distributed across multiple devices. Each device computes its layers and passes the result to the next device until the entire inference is complete. (c) *Galaxy* (Ye et al., 2024) combines tensor and sequence parallelism and overlaps communication and computation to accelerate inference. They all run in FP32 mode.

**Run larger models with lower latency and memory usage.** As shown in Figure 6, a limited memory on a single device makes it challenging to run even 3B models in a standalone mode. MP addresses this by the collaboration of 8 devices, allowing models up to 13B, but suffers from high latency due to pipeline bubbles. Galaxy tries to reduce such latency by combining tensor and sequence parallelism. However, in Section 3.2, we concluded that the network bandwidth was no longer the issue, and the real problem is the link latency. Galaxy's use of a ring algorithm for reducescatter and allgather forces each link to be traversed at least 14 times. This causes high link latency and outweighs the benefits of parallel computing, ultimately resulting in a higher token latency than MP. In contrast, TPI-LLM adopts a star-based allreduce algorithm, minimizing hops and cumulative link latency. Combined with the blocking-free memory scheduler, TPI-LLM delivers significantly lower token latency and memory footprint, even with larger 70B models.

## 4.4 REAL CASE STUDY

In this study, we used 4 laptops with different CPU architectures and memory capacities, connected via a local Wi-Fi router. The testbed and configurations are detailed in Appendix A.10. Macbook Pro was used by default. Due to the lack of CUDA, all computations were performed in full precision. As shown in Table 3, Transformers loaded the entire model into the CPU memory, and when memory was insufficient, the operating system offloaded data to the swap. This frequent swap exchange significantly increased TTFT and token latency, even for smaller 3B models. As the model size grows, the swap space overflowed, finally leading to OOM errors. As a more efficient alternative, Accelerate (Gugger et al., 2022) instantly loads layer weights only when required for the

Table 3: Comparison of Transformers, Accelerate, Transformers+MS, and TPI-LLM on 4 laptops.

| Model (FP32) | Transformers | | Accelerate | | Transformers + MS | | TPI-LLM | |
|---|---|---|---|---|---|---|---|---|
| | TTFT (s) | Latency (s/token) | TTFT (s) | Latency (s/token) | TTFT (s) | Latency (s/token) | TTFT (s) | Latency (s/token) |
| Llama 2-3B | 61 | 30 | 24 | 16 | 4 | 3 | **2.5** | **2** |
| Llama 2-7B | 115 | 56 | 30 | 26 | 13 | 8 | **6** | **5** |
| Llama 2-13B | OOM | OOM | OOM | OOM | 22 | 18 | **10** | **9** |
| Llama 3.1-8B | 133 | 65 | 37 | 31 | 20 | 12 | **11** | **8** |
| Yi-34B | OOM | OOM | OOM | OOM | 185 | 55 | **33** | **29** |

computation and reduces unnecessary data I/O. While it speeds up inference, due to implementation flaws on disk offloading, it still requires loading full weights before splitting and offloading them to disk. This results in OOM errors when the model size reaches 13B.

**TPI-LLM stands out in TTFT, token latency, and model size.** Our memory scheduler (Transformers+MS) outperforms Transformers and Accelerate in both TTFT and token latency across all model sizes. This is because our memory scheduler employs a sliding window mechanism, where a daemon thread asynchronously preloads the weights needed for upcoming computations. By overlapping data I/O with computations and communications, the scheduler avoids delays caused by disk I/O blocks, ensuring smoother and faster inference. To further speed up inference, we integrate the computing power of 4 laptops to serve TPI-LLM. By distributing the computational load across 4 laptops, the reduction in computing time far exceeds communication delays, so both TTFT and token latency are further reduced. The results from using 3 laptops are shown in Appendix A.11, indicating a slightly higher latency due to reduced parallelism.

## 5 CONCLUSION

In this paper, we concluded that tensor parallelism can be more effective than pipeline parallelism on low-resource devices, and presented a compute- and memory-efficient tensor parallel inference system, named TPI-LLM, to serve 70B-scale LLMs. TPI-LLM is designed with user prompt and generated sequence privacy in mind, by keeping sensitive raw data local in the users' devices. It leverages a sliding window memory scheduler to dynamically manage layer weights during inference with disk I/O latency overlapped by onging computations and communications, allowing larger models to run smoothly on devices with very limited memory. Our analysis showed that link latency, not bandwidth, emerges as the main issue, so TPI-LLM implements a star-based allreduce algorithm, rather than the commonly used ring- and tree-based algorithms. Through extensive experiments on emulated and real testbeds, TPI-LLM demonstrated significantly lower TTFT, token latency, and peak memory footprint compared to Transformers, Accelerate, Galaxy, and enabled serving larger-scale LLMs such as Yi-34B and Llama 2/3/3.1-70B on low-memory devices.

### REPRODUCIBILITY

We have made efforts to ensure reproducibility by providing the source code at `https://anonymous.4open.science/r/tpi-llm`, with a detailed README for guidance included. To ease the use, a prebuilt Docker image is also provided. Key experimental setups are given in Section 4 of the paper.

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

# A APPENDIX

## A.1 PROOF OF PROPOSITION 2

In conventional data parallel systems, each device sends several gigabytes of data, putting significant pressure on network bandwidth. This makes data transfer latency a major concern, while link latency becomes negligible. Then, tree and ring-based algorithms are introduced to optimize the data transfer. However, they do not apply to our case. In TPI-LLM, each device only sends a small amount of data, usually just tens of kilobytes. This tiny data size does not strain the network, so data transfer latency is minimal. Instead, in edge networks where wireless communication causes higher transmission delays, link latency becomes more significant than data transfer latency. As a result, the commonly used tree and ring-based allreduce algorithms are less effective.

Let us consider 1 master and 2 workers connected via a router. In Figure 7, we compare the traffic models of star, tree, and ring-based algorithms. In star-based allreduce, worker 1 sends data directly to the master via the router, and the allreduce latency (includes reduce and broadcast) is $t_{\texttt{star}} = 2(t_{\texttt{data}} + t_{\texttt{link}}) + t_{\texttt{barrier}} + t_{\texttt{aggr}}$. In this model, the router only forwards data packets.

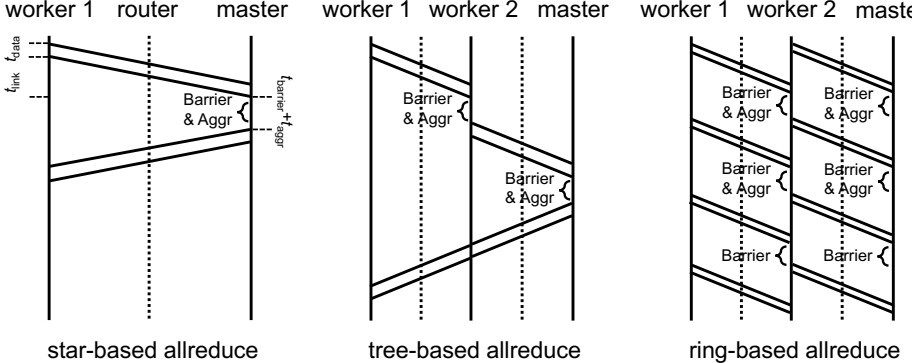

Figure 7: Comparison of traffic models for star, tree, and ring-based allreduce algorithms.

In tree-based allreduce, data from worker 1 must first go through worker 2 before reaching the master, so there are 2 hops involved. In this process, worker 1 sends its data to worker 2, which aggregates it and forwards the result to the master. Once the global aggregation is complete, the final result is broadcast back to all workers. The total time for this process is $t_{\texttt{tree}} = 3t_{\texttt{data}} + 4t_{\texttt{link}} + 2t_{\texttt{barrier}} + 2t_{\texttt{aggr}}$.

In ring-based allreduce, each device communicates directly with its neighbors in a ring topology. Data is divided and sent in a sequence around the ring, with each device receiving, aggregating, and passing the data to the next device. Unlike star or tree-based methods, there is no central device, and data flows continuously between the devices. The total time for the ring-based allreduce is $t_{\texttt{ring}} = \frac{4}{3}t_{\texttt{data}} + 4t_{\texttt{link}} + 3t_{\texttt{barrier}} + \frac{2}{3}t_{\texttt{aggr}}$.

Assume that all devices are homogeneous, i.e., $t_{\texttt{barrier}} \approx 0$, and $t_{\texttt{data}} \approx 0, t_{\texttt{aggr}} \approx 0$ because the data size is very small. Then we have latencies simplified as follows:

$$t_{\texttt{star}} = 2t_{\texttt{link}} < t_{\texttt{tree}} = t_{\texttt{ring}} = 4t_{\texttt{link}}. \tag{11}$$

Thus, the star-based allreduce is the most efficient method because it minimizes link latency.

## A.2 A SIMPLE-TO-USE MEMORY SCHEDULER

In our implementation, a context manager is used to ensure that the required block weights are loaded correctly and unload the used weights to free up memory for subsequent blocks. This simplifies the deployment of large-scale LLMs on low-memory edge devices, requiring just one additional line of code:

```
1  with memory_manager.wait_and_release(f"self_attn.0"):
2      hidden_states = self_attn(hidden_states)
```

### A.3 PROOF OF PROPOSITION 3

We start with the first attention block and end with the final FFN block.

*Time slot 1 (attention computation):* In this initialization step, $W_{\text{attn}}^1$ must be loaded before computing the first attention block, taking $\tau_{\text{attn}} + t_{\text{attn}}$. During the computation time $t_{\text{attn}}$, the next FFN weights, $W_{\text{ffn}}^1$, are loading in parallel.

*Time slot 2 (allreduce):* The attention block is followed by allreduce communication, which takes $t_{\text{all\_reduce}}$, with the next FFN weights, $W_{\text{ffn}}^1$, loading in parallel.

*Time slot 3 (FFN computation):* By this time, the FFN weights $W_{\text{ffn}}^1$ should be fully loaded. If not, the computation must wait for loading to complete. Let $t' = t_{\text{attn}} + t_{\text{all\_reduce}} - \tau_{\text{ffn}}$, if $t' \geq 0$, no blocking occurs; otherwise, the computation is delayed by $|t'|$. Once loaded, compute the FFN block in $t_{\text{ffn}}$.

During this time slot, the waiting, computation of the current FFN block and the weight loading of the next attention block occur simultaneously. By the time the current FFN block finishes, the next attention block's weights $W_{\text{attn}}^2$ have been loading for $\max\{0, t_{\text{attn}} + t_{\text{all\_reduce}} - \tau_{\text{ffn}}\} + t_{\text{ffn}}$.

*Time slot 4 (allreduce):* The FFN block is followed by allreduce communication, which takes $t_{\text{all\_reduce}}$, with the next attention weights, $W_{\text{attn}}^2$, loading in parallel.

*Time slot 5 (attention computation):* Ensure that the attention weights $W_{\text{attn}}^2$ are fully loaded. Let $t' = \max\{0, t_{\text{attn}} + t_{\text{all\_reduce}} - \tau_{\text{ffn}}\} + t_{\text{ffn}} + t_{\text{all\_reduce}} - \tau_{\text{attn}}$. If $t' \geq 0$, the computation proceeds without blocking. Then, $W_{\text{attn}}^2$ is computed in $t_{\text{attn}}$, and the next FFN weights $W_{\text{ffn}}^2$ have been loading for $\max\{0, \max\{0, t_{\text{attn}} + t_{\text{all\_reduce}} - \tau_{\text{ffn}}\} + t_{\text{ffn}} + t_{\text{all\_reduce}} - \tau_{\text{attn}}\} + t_{\text{attn}}$.

*Time slot 6 (allreduce):* The allreduce communication takes $t_{\text{all\_reduce}}$, while the next FFN weights $W_{\text{ffn}}^2$ are loading in parallel.

*Time slot 7 (FFN computation):* Ensure that the FFN weights $W_{\text{ffn}}^2$ are fully loaded. Let $t' = \max\{0, \max\{0, t_{\text{attn}} + t_{\text{all\_reduce}} - \tau_{\text{ffn}}\} + t_{\text{ffn}} + t_{\text{all\_reduce}} - \tau_{\text{attn}}\} + t_{\text{attn}} + t_{\text{all\_reduce}} - \tau_{\text{ffn}}$. If $t' \geq 0$, the computation proceeds without blocking.

This process repeats, until the generation task is finished.

For the system to reach a steady state where computation is not blocked by weight loading at any time, the following conditions must hold.

*Case 1:* $t_{\text{attn}} + t_{\text{all\_reduce}} - \tau_{\text{ffn}} \geq 0$.

$$\text{Time slot 3 } (l=1): \quad t_{\text{attn}} + t_{\text{all\_reduce}} - \tau_{\text{ffn}} \geq 0, \tag{12}$$

$$\text{Time slot 5 } (l=1): \quad t_{\text{attn}} + t_{\text{ffn}} + 2t_{\text{all\_reduce}} - \tau_{\text{ffn}} - \tau_{\text{attn}} \geq 0, \tag{13}$$

$$\text{Time slot 7 } (l=2): \quad 2t_{\text{attn}} + t_{\text{ffn}} + 3t_{\text{all\_reduce}} - 2\tau_{\text{ffn}} - \tau_{\text{attn}} \geq 0, \tag{14}$$

$$\text{Time slot 9 } (l=2): \quad 2t_{\text{attn}} + 2t_{\text{ffn}} + 4t_{\text{all\_reduce}} - 2\tau_{\text{ffn}} - 2\tau_{\text{attn}} \geq 0. \tag{15}$$

We repeat these conditions and derive the following patterns.

$$t_{\text{attn}} + t_{\text{ffn}} + 2t_{\text{all\_reduce}} \geq \tau_{\text{ffn}} + \tau_{\text{attn}}, \tag{16}$$

$$l \cdot t_{\text{attn}} + (l-1) \cdot t_{\text{ffn}} + (2l-1) \cdot t_{\text{all\_reduce}} \geq l \cdot \tau_{\text{ffn}} + (l-1) \cdot \tau_{\text{attn}}. \tag{17}$$

*Case 2:* $t_{\text{attn}} + t_{\text{all\_reduce}} - \tau_{\text{ffn}} < 0$.

$$\text{Time slot 3 } (l=1): \quad t_{\text{attn}} + t_{\text{all\_reduce}} - \tau_{\text{ffn}} < 0, \tag{18}$$

$$\text{Time slot 5 } (l=1): \quad t_{\text{ffn}} + t_{\text{all\_reduce}} - \tau_{\text{attn}} \geq 0, \tag{19}$$

$$\text{Time slot 7 } (l=2): \quad t_{\text{attn}} + t_{\text{ffn}} + 2t_{\text{all\_reduce}} - \tau_{\text{attn}} - \tau_{\text{ffn}} \geq 0, \tag{20}$$

$$\text{Time slot 9 } (l=2): \quad t_{\text{attn}} + 2t_{\text{ffn}} + 3t_{\text{all\_reduce}} - 2\tau_{\text{attn}} - \tau_{\text{ffn}} \geq 0, \tag{21}$$

$$\text{Time slot 11 } (l=3): \quad 2t_{\text{attn}} + 2t_{\text{ffn}} + 4t_{\text{all\_reduce}} - 2\tau_{\text{attn}} - 2\tau_{\text{ffn}} \geq 0. \tag{22}$$

Similarly, repeat these conditions and derive the following patterns.

$$t_{\texttt{attn}} + t_{\texttt{ffn}} + 2t_{\texttt{all\_reduce}} \geq \tau_{\texttt{ffn}} + \tau_{\texttt{attn}}, \tag{23}$$

$$(l-1) \cdot t_{\texttt{attn}} + l \cdot t_{\texttt{ffn}} + (2l-1) \cdot t_{\texttt{all\_reduce}} \geq (l-1) \cdot \tau_{\texttt{ffn}} + l \cdot \tau_{\texttt{attn}}. \tag{24}$$

Thus, the proposition is proved.

### A.4    PROOF OF PROPOSITION 4

Let $\alpha = l \cdot t_{\texttt{attn}} + (l-1) \cdot t_{\texttt{ffn}} + (2l-1) \cdot t_{\texttt{all\_reduce}} - l \cdot \tau_{\texttt{ffn}} - (l-1) \cdot \tau_{\texttt{attn}} > 0$, and we derive the following inequality from inequality (16):

$$l \cdot t_{\texttt{attn}} + l \cdot t_{\texttt{ffn}} + 2l \cdot t_{\texttt{all\_reduce}} - l \cdot \tau_{\texttt{ffn}} - l \cdot \tau_{\texttt{attn}} > 0. \tag{25}$$

By substituting $\alpha$ into this inequality, we have $\alpha + t_{\texttt{ffn}} + t_{\texttt{all\_reduce}} - \tau_{\texttt{attn}} > 0$. Let $\alpha > 0 > \tau_{\texttt{attn}} - t_{\texttt{ffn}} - t_{\texttt{all\_reduce}}$, we obtain the first condition:

$$t_{\texttt{ffn}} + t_{\texttt{all\_reduce}} > \tau_{\texttt{attn}}. \tag{26}$$

Let $\beta = t_{\texttt{ffn}} + t_{\texttt{all\_reduce}} - \tau_{\texttt{attn}} > 0$, and substitute $\beta$ into inequality (16), then we have $\beta + t_{\texttt{attn}} + t_{\texttt{all\_reduce}} - \tau_{\texttt{ffn}} > 0$. Let $\beta > 0 > \tau_{\texttt{ffn}} - t_{\texttt{attn}} - t_{\texttt{all\_reduce}}$, we obtain the second condition:

$$t_{\texttt{attn}} + t_{\texttt{all\_reduce}} > \tau_{\texttt{ffn}}. \tag{27}$$

Thus, the proposition is proved.

### A.5    PROOF OF PROPOSITION 5

In this section, we analyze the peak memory footprint on both the master and worker nodes to estimate the largest model size that our memory scheduler can handle.

Let us use the Llama model as an example, assume the vocabulary size is $v$, hidden size is $h$, number of attention heads is $a$, number of key-value heads is $b$, and intermediate size is $s$. Let $\boldsymbol{p} = [p_1, p_2, \cdots, p_n]$ be a vector representing the proportion of parameters handled by $n$ devices, and $w$ be the window size of the memory scheduler. Following the block definition in Figure 2, the parameter counts for each block are detailed in Table 4:

Table 4: Parameter counts for the main blocks (e.g., $n = 4$, $p_i = 0.25$, Llama 2-7B).

| Block | Parameters | Block Size |
|---|:---:|:---:|
| Preprocess | $hv$ | 500 MB |
| Attention | $2(a+b)h^2 p_i/a + h$ | 64 MB |
| FFN | $3hsp_i + h$ | 129 MB |
| Postprocess | $hv + h$ | 500 MB |

The memory footprint is affected by parameters, activation storage, temporary tensors, memory management, and caching, making precise quantification challenging. To estimate peak memory, we apply an empirical rule: multiply the parameter size by a scaling factor $\gamma$.



Figure 8: Illustration of the memory window at the peak memory footprint.

From the memory window at the peak memory footprint shown in Figure 8, we can derive the following equations.

$$M_{\texttt{master}} = \gamma \times \begin{cases} hv + h, & \text{if } w = 1 \\ 2hv + h, & \text{if } w = 2 \\ 2hv + h + \left\lfloor \frac{w-2}{2} \right\rfloor \left( 2(1 + \frac{b}{a})h^2 p_i + h \right) + \left\lfloor \frac{w-1}{2} \right\rfloor (3hsp_i + h), & \text{if } w \geq 3 \end{cases}$$

For any worker node, the memory footprint does not include the preprocess and postprocess blocks. Therefore, the peak memory footprint $M_{\texttt{worker}}$ can be expressed as:

$$M_{\texttt{worker}} = \gamma \times \left( \left\lfloor \frac{w}{2} \right\rfloor \left( 2(1 + \frac{b}{a})h^2 p_i + h \right) + \left\lfloor \frac{w+1}{2} \right\rfloor (3hsp_i + h) \right).$$

Thus, the proposition is proved.

### A.6 PROOF OF PROPOSITION 6

When the memory scheduler reaches a steady state, the overlap between computation, communication, and disk I/O is optimized, ensuring that weights are always pre-loaded before they are needed for computations. However, if disk I/O becomes a bottleneck and disrupts the steady state (e.g., due to high disk latency), the scheduler must adapt by selectively retaining certain blocks in memory to reduce disk access frequency.

In our preliminary experiments, we measured $t_{\texttt{attn}} = 11$ ms, $t_{\texttt{ffn}} = 17$ ms, $\tau_{\texttt{attn}} = 18$ ms, $\tau_{\texttt{ffn}} = 30$ ms, and observed that FFN blocks generally exhibit higher computation and weight loading latency. By retaining some FFN blocks in memory, we can reduce the need to reload large weights.

Let the memory scheduler retain one FFN block in memory every $T$ FFN blocks, and

$$\mathbb{I}_{\{l=1+kT\}} = \begin{cases} 1, & \text{if } l = 1 + kT \text{ and } k \in \mathbb{Z} \geq 0, \\ 0, & \text{otherwise.} \end{cases}$$

Similar to the analysis in Appendix A.3, we have

*Time slot 3 ($l = 1$):* $\quad t_{\texttt{attn}} + t_{\texttt{all\_reduce}} - (1 - \mathbb{I}_{\{l=1+kT\}})\tau_{\texttt{ffn}} \geq 0,$

*Time slot 5 ($l = 1$):* $\quad t_{\texttt{attn}} + t_{\texttt{ffn}} + 2t_{\texttt{all\_reduce}} - (1 - \mathbb{I}_{\{l=1+kT\}})\tau_{\texttt{ffn}} - \tau_{\texttt{attn}} \geq 0,$

*Time slot 7 ($l = 2$):* $\quad 2t_{\texttt{attn}} + t_{\texttt{ffn}} + 3t_{\texttt{all\_reduce}} - \sum_{i=1}^{2}(1 - \mathbb{I}_{\{i=1+kT\}})\tau_{\texttt{ffn}} - \tau_{\texttt{attn}} \geq 0,$

*Time slot 9 ($l = 2$):* $\quad 2t_{\texttt{attn}} + 2t_{\texttt{ffn}} + 4t_{\texttt{all\_reduce}} - \sum_{i=1}^{2}(1 - \mathbb{I}_{\{i=1+kT\}})\tau_{\texttt{ffn}} - 2\tau_{\texttt{attn}} \geq 0,$

*Time slot 11 ($l = 3$):* $\quad 3t_{\texttt{attn}} + 2t_{\texttt{ffn}} + 5t_{\texttt{all\_reduce}} - \sum_{i=1}^{3}(1 - \mathbb{I}_{\{i=1+kT\}})\tau_{\texttt{ffn}} - 2\tau_{\texttt{attn}} \geq 0.$

By repeating these conditions, we derive the following patterns:

$$l \cdot t_{\texttt{attn}} + l \cdot t_{\texttt{ffn}} + 2l \cdot t_{\texttt{all\_reduce}} - \sum_{i=1}^{l}(1 - \mathbb{I}_{\{i=1+kT\}})\tau_{\texttt{ffn}} - l \cdot \tau_{\texttt{attn}} \geq 0,$$

$$l \cdot t_{\texttt{attn}} + (l-1) \cdot t_{\texttt{ffn}} + (2l-1) \cdot t_{\texttt{all\_reduce}} - \sum_{i=1}^{l}(1 - \mathbb{I}_{\{i=1+kT\}})\tau_{\texttt{ffn}} - (l-1) \cdot \tau_{\texttt{attn}} \geq 0.$$

Since $\sum_{i=1}^{l} \mathbb{I}_{\{i=1+kT\}} = \left\lceil \frac{l}{T} \right\rceil$, we have

$$l \cdot t_{\texttt{attn}} + l \cdot t_{\texttt{ffn}} + 2l \cdot t_{\texttt{all\_reduce}} \geq (l - \left\lceil \frac{l}{T} \right\rceil) \cdot \tau_{\texttt{ffn}} + l \cdot \tau_{\texttt{attn}},$$

$$l \cdot t_{\texttt{attn}} + (l-1) \cdot t_{\texttt{ffn}} + (2l-1) \cdot t_{\texttt{all\_reduce}} \geq (l - \left\lceil \frac{l}{T} \right\rceil) \cdot \tau_{\texttt{ffn}} + (l-1) \cdot \tau_{\texttt{attn}}.$$

Thus, the proposition is proved.

## A.7 KLONET TESTBED

One of our testbed, as shown in Figure 9, was built upon Klonet (Ma et al., 2024) to create an edge network environment. Klonet is a network emulation platform designed to support the development and testing of new network protocols and applications in a realistic environment. It can emulate various network scenarios, such as wireless, mobile, satellite, and optical networks, and provide fine-grained control over the network parameters, such as bandwidth, delay, jitter, and packet loss. It can also integrate with real devices and applications, such as routers, switches, sensors, and smartphones, to create hybrid network experiments. Klonet is based on the Linux operating system and uses virtualization and containerization technologies to create isolated network nodes and links. It provides both GUI and CLI to help users configure and manage their network experiments.

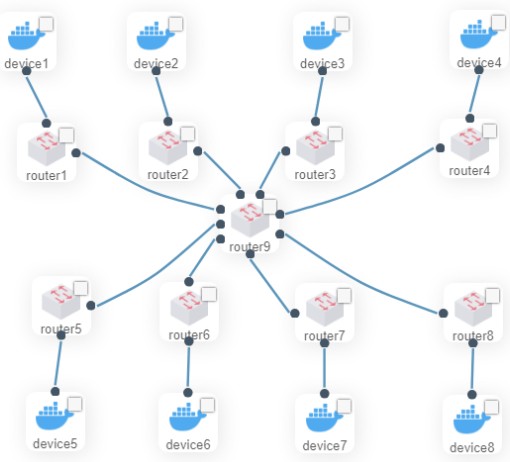

Figure 9: Testbed built upon Klonet.

This testbed includes 8 user devices (devices 1 to 8), 8 home gateways (routers 1 to 8), and 1 core router (router 9). User devices connect to their home gateways via wired or wireless connections, and these home gateways are interconnected through routers or switches in the edge network. This topology reflects real-world household network interconnections. In addition, the CPU cores, memory, swap limits, bandwidth, and latency settings in Section 4 are based on measurements from the authors' edge network.

## A.8 CONFIGURATIONS OF THE USED MODELS

Table 5: Configurations of the used Llama models.

| Model (FP32) | Layers | Params | Hidden Size | Heads | KV Heads | Required Mem |
|---|---|---|---|---|---|---|
| Llama 2-3B | 26 | 3 billion | 3200 | 32 | – | 14 GB |
| Llama 2-7B | 32 | 7 billion | 4096 | 32 | – | 26 GB |
| Llama 2-13B | 40 | 13 billion | 5120 | 40 | – | 50 GB |
| Llama 2-70B | 80 | 70 billion | 8192 | 64 | 8 | 257 GB |
| Llama 3.1-8B | 32 | 8 billion | 4096 | 32 | 8 | 31 GB |
| Llama 3.1-70B | 80 | 70 billion | 8192 | 64 | 8 | 266 GB |
| Yi-34B | 60 | 34 billion | 7168 | 56 | 8 | 130 GB |

## A.9 PEAK MEMORY FOOTPRINT WITH MEMORY WINDOW SIZE 4

Table 6: Peak memory footprint per device with the memory window size set to 4.

| | Memory Scheduler Disabled (GB) | | | | Memory Scheduler Enabled (GB) | | | |
|---|---|---|---|---|---|---|---|---|
| Model (FP32) | $N=2$ | $N=4$ | $N=6$ | $N=8$ | $N=2$ | $N=4$ | $N=6$ | $N=8$ |
| Llama 2-3B | 7.3 | 4.3 | 3.2 | 2.8 | 1.7 | 1.5 | 1.5 | 1.5 |
| Llama 2-7B | 13.7 | 7.7 | 5.5 | 4.5 | 2.4 | 2.1 | 1.8 | 1.8 |
| Llama 2-13B | 25.8 | 13.9 | 9.7 | 8.0 | 2.8 | 2.5 | 2.3 | 2.2 |
| Llama 2-70B | 129.9 | 66.5 | 46.7 | 35.0 | 4.5 | 3.1 | 3.1 | 3.1 |
| Llama 3.1-8B | 18.4 | 11.8 | 9.4 | 8.5 | 6.3 | 5.8 | 5.6 | 5.5 |
| Llama 3.1-70B | 137.8 | 74.0 | 51.4 | 42.5 | 10.8 | 10.5 | 11.5 | 11.4 |
| Yi-34B | 67 | 36.4 | 23.9 | 20.4 | 6.0 | 5.6 | 5.3 | 5.2 |

## A.10 REAL TESTBED AND CONFIGURATIONS

The real testbed consists of 4 laptops, all connected via a local Wi-Fi router, as shown in Figure 10. Table 7 details the hardware and network configurations of these laptops. In this case study, the laptop in the lower right serves as the master, while the other three laptops act as workers. The workers are connected to the master, and the master is currently generating the output sequence. The generated sequence is identical to that of single-server inference.

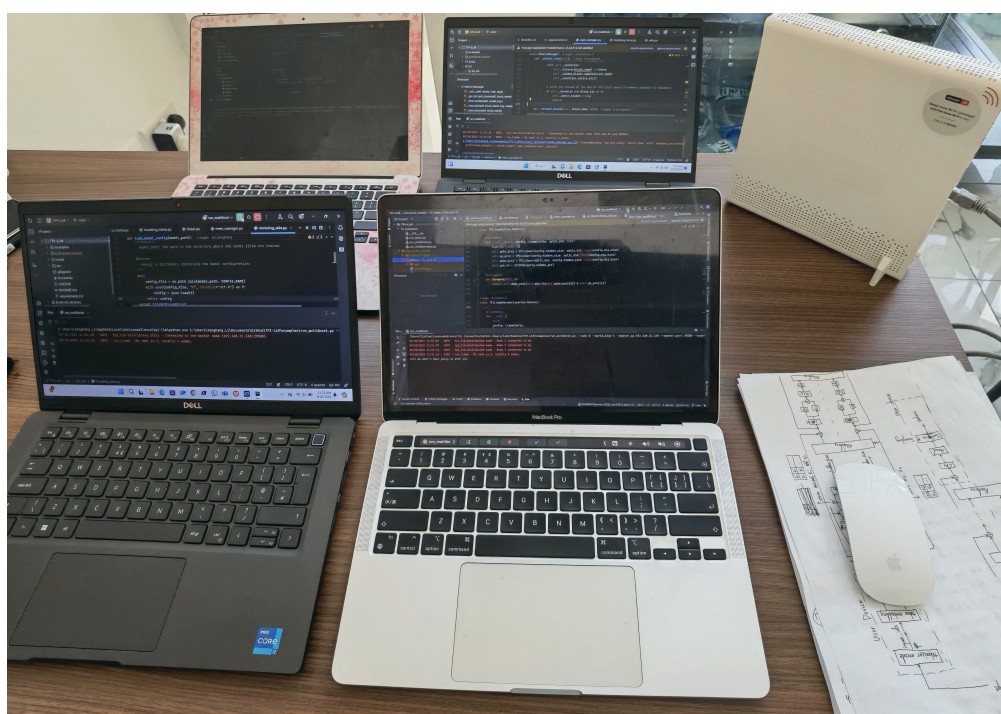

Figure 10: A real testbed composed of 4 laptops connected via local Wi-Fi.

Table 7: Hardware and network configurations of the laptops.

| Device | CPU Model | Cores | Memory | Bandwidth | Latency | CUDA | Number |
|---|---|---|---|---|---|---|---|
| Mac Pro | Apple M1 | 8 | 8 GB | 510 Mbps | 5 ms | No | 1 |
| Mac Air | Intel Core i5 | 4 | 8 GB | 320 Mbps | 7 ms | No | 1 |
| Dell | Intel i7-1165G7 | 8 | 16 GB | 610 Mbps | 3 ms | No | 2 |

## A.11 Case study with 3 laptops

In this case, only 3 out of the 4 laptops are used: one MacBook Pro, one MacBook Air, and one Dell laptop. The results are given in Table 8, indicating a slightly higher latency due to reduced parallelism.

Table 8: Comparison of Transformers, Accelerate, Transformers+MS, and TPI-LLM on 3 laptops.

| Model (FP32) | Transformers | | Accelerate | | Transformers + MS | | TPI-LLM | |
|---|---|---|---|---|---|---|---|---|
| | TTFT (s) | Latency (s/token) | TTFT (s) | Latency (s/token) | TTFT (s) | Latency (s/token) | TTFT (s) | Latency (s/token) |
| Llama 2-3B | 61 | 30 | 24 | 16 | 4 | 3 | **3** | **2** |
| Llama 2-7B | 115 | 56 | 30 | 26 | 13 | 8 | **7** | **6** |
| Llama 2-13B | OOM | OOM | OOM | OOM | 22 | 18 | **14** | **12** |
| Llama 3.1-8B | 133 | 65 | 37 | 31 | 20 | 12 | **13** | **9** |
| Yi-34B | OOM | OOM | OOM | OOM | 185 | 55 | **48** | **41** |