# OpenReview forum: "TPI-LLM: Serving 70B-scale LLMs Efficiently on Low-resource Edge Devices"
_ICLR.cc/2025/Conference — ICLR 2025 Conference Withdrawn Submission_

### Official Review · Reviewer_jo28 · 2024-11-03

**Soundness:** 3
**Presentation:** 3
**Contribution:** 2
**Rating:** 5
**Confidence:** 4

**Summary:**

The paper proposes TPI-LLM, a tensor parallelism-based distributed LLM inference system tailored to low-resource edge devices. Unlike inference tasks on servers, TPI-LLM keeps user data securely on user (edge) devices during inference. By analyzing the network latency bottleneck and memory limits, TPI-LLM leverages a star-based all-reduce algorithm to facilitate distributed inference and employs a sliding window-based memory scheduler to reduce inference memory footprints. Experiments on both emulations and real-world testbeds show that TPI-LLM outperforms existing baselines by providing lower TTFT, token latency, and peak memory footprints.

**Strengths:**

1 - The paper provides extensive theoretical analysis.

2 - The proposed approach is evaluated in both emulations and real-world testbeds against state-of-the-art baselines.

3 - The performance is significant, especially on edge devices with limited resources.

**Weaknesses:**

1 - In Section 2, Q1 is somewhat ambiguous. First, isn't tensor parallelism already a form of model parallelism? And second, even on low-resource edge devices, can we combine and use both parallelism techniques instead of simply abandoning one?

2 - In Section 3.2, the example network topology here is star-based (Appendix A.7). Given this star-based topology, a star-based all-reduce scheme indeed would be most efficient. Is this a common network topology for all edge scenarios?

3 - Figure 4 may need further detailed descriptions. The sliding window is not very clear in the figure. For example, in Time 7 and 8, why would you prefetch Blocks 6, 7, and 8 so early when it's still far away from actually using them? Isn't that prefetching too early causing memory waste?

4 - Note that the layer-wise parameter offloading is not new. Many popular frameworks, such as DeepSpeed-Inference, Mixtral Offload, and Llama.cpp, support this offloading scheme. How does the proposed TPI-LLM differ from existing offloading techniques?

5 - Evaluation lacks sensitivity analysis on the memory sliding window size. Why would you pick a window size of 2?

**Questions:**

Please see questions from weaknesses.

---

### Official Review · Reviewer_3Ruh · 2024-11-03

**Soundness:** 2
**Presentation:** 2
**Contribution:** 3
**Rating:** 5
**Confidence:** 4

**Summary:**

The paper proposes a new framework called TPI-LLM for model serving on low resource edge devices using tensor parallelism and memory scheduling. The proposed frameworks shows better performance over the SOTA frameworks in terms of latency to predict the first token and the overall token latency.

**Strengths:**

Following are the strengths:
- The paper is well-written, easy to follow and understand the concepts presented.
- The paper tries to addresses the critical problem of LLM inference on the edge devices.
- The paper discusses the existing frameworks in this space and positions within the body of knowledge.
- Achieving the faster TTFT latency using 3k LOC and for large models is always interesting and of use for broader audience.
- Although the paper uses multiple devices connected in the same network (Wi-Fi/cable connected), it is understandable that there are use-cases where a house will have multiple edge devices all of which might operate in-tandem, there maybe business orgs that have edge devices that don't have permission issues to run LLMs on multiple of such devices. For such use-cases this is a significant contribution.
- The paper proposed an easy to use technique of overlaying the data fetch during communication step in LLMs in the proposed sliding window strategy.

**Weaknesses:**

Please follow the questions section, there is a cohesive list of weaknesses and the corresponding questions that are to be addressed.

**Questions:**

- This technique might also be useful even in the cloud server settings, especially when there is not enough GPUs, can the sliding window memory management can help avoid the OOMs. Any thoughts in that direction or future recommendations maybe?

However, following are the concerns/questions.

### Major concerns
- The proposed framework has two major operational changes that are applied on the pre-trained LLM, that are 1) distribution of attention heads across nodes, ii) the memory management through the sliding window approach. Given these two changes, the paper does not discuss the performance implications with and without the proposed framework. It is important to guarantee that the performance remains same.
  - Given that, it is recommended to show that the performance (atleast in the best case settings on atleast one model) remains unchanged with or without the TPI-LLM.
- Figure 5 is more concerning in the following perspectives.
  - No impact on bandwidth: There is no surprise on the lack of impact of increasing the bandwidth since the sliding window size is defaulted to 2. However, we can only learn about the impact by increasing the window size. There is no such ablation in the paper that shows a best combination of the number of devices, maximum possible sliding window size, bandwidth, available memory on each of the devices.
    - Recommendation is to provide a thorough feasibility study on the combination of the above variables to clearly understand the impact of the bandwidth. In fact on that note, there is no clear analysis on the maximum possible sliding window size for a given hardware configuration of the master/worker nodes. Therefore, the feasibility study can be preceded by maximum window size in order to limit the number of combinations to be studied.
  - Increasing the number of devices/cores reduces the token latency (from first two sub-figures in Figure5) is not a true statement and is really vague. Those plots are shown simply for 8 devices, if the number of devices are kept increasing, after a point we see diminishing returns of parallelism. That is where the communication dominates and hence to diminishing returns. Without having proper study `increasing the devices reduces the latency` are not valid claims.
    - The recommendation is to conduct a quick roofline analysis to substantiate the claims or remove the controversial parts.
- There are a few limitations on what the proposed framework can offer. They are as follows.
 - There is a security and privacy concern, this framework should not run any device connected in the same Wi-Fi network unless there is a prior consent. It is not stated in addressed in the paper and hence please clearly state the limitation or the constraints under which the framework can operate.
 - The star configuration can not scale to large number of nodes/devices. It probably can be extended to scale in a hierarchical-star configuration etc, but that is not the scope of the paper and hence this needs to be state clearly as a limitation. There are real-time use-cases in the resource constrained edge scenarios where the number of devices is high, which leads to failures of a single master node in star config.
 - There is probably an unstated assumption in the paper that the data gets generated on the master or stays centrally on the master node/device. However, there is a high chance of the worker nodes/devices having user specific data. It appears that the proposed framework does not handle data that is generated on all the other devices. If that is not the case, please clarify how that data is handled on each of the workers? Otherwise, this limitation should be stated.

### Minor concerns
- In Figure 4, Time steps 7 and 8 have the same memory window, why is it the case, it is a bit confusing to understand. Is the memory peaked and hence the window does not slide or something else? Please provide clarifications or amend the figure to make it clear.
- Tables 1 & 2 provide comparisons for TTFT, latency and memory usage, etc between with and without the use of memory scheduler. However, when the memory scheduler is disabled, it is not clear how those numbers are attained. How were they measured? by using accelerate, vanilla Klonet or Galaxy or llama.cpp or others?
  - Please add those details on how the stats were measured and the underlying frameworks. Ideally benchmark comparisons against the best possible frameworks/methods is a common practice?
- Section 4.3 states the `comparison with benchmarks` Ideally those (a. Standalone, b. Model Parallelism (MP) and c. Galaxy) are SOTA methods/frameworks (in this case). Are they not? Why are they called benchmarks? Clearly there is benchmarking of TPI-LLM against those other things.
 - Recommendation is to please rephrase in order to convey the message so that the confusing the reader can be avoided.

---

### Official Review · Reviewer_rKWH · 2024-11-04

**Soundness:** 2
**Presentation:** 2
**Contribution:** 2
**Rating:** 5
**Confidence:** 4

**Summary:**

The paper presents a novel tensor parallel inference system named TPI-LLM, designed to efficiently serve large-scale language models (LLMs) on low-resource edge devices. The system addresses the challenges of limited computing power, memory, and bandwidth on edge devices by introducing a sliding window memory scheduler that dynamically manages layer weights during inference, overlapping disk I/O latency with computation and communication. This approach allows larger models to run smoothly on memory-limited devices while keeping sensitive raw data local to the user's devices, enhancing privacy.

The key contributions of the paper are as follows:

1. Tensor Parallelism on Edge Devices: The paper argues for the effectiveness of tensor parallelism over pipeline parallelism on low-resource edge devices and presents TPI-LLM as a compute- and memory-efficient solution for serving 70B-scale models.

2. Sliding Window Memory Scheduler: A memory scheduler is introduced to asynchronously load and unload layer weights, which enables the inference of larger models on devices with limited memory by overlapping disk I/O with computations and communications.

3. Star-Based AllReduce Algorithm: The paper identifies link latency as the main issue in communication and implements a star-based allreduce algorithm to reduce latency, outperforming ring- and tree-based methods commonly used in high-latency networks.

**Strengths:**

This paper addresses the emerging challenge of deploying large language models (LLMs) on edge devices, which is a novel and increasingly relevant problem in the era of edge computing and privacy concerns. It proposes a new approach to tensor parallelism that is specifically tailored for low-resource environments, which is an original contribution to the field.

The authors combine concepts from distributed computing, memory management, and parallelism to create a system that is both memory and compute-efficient. The sliding window memory scheduler and the star-based allreduce algorithm are creative solutions that address specific pain points in edge device inference.

The work has significant implications for the future of edge computing, as it enables the deployment of LLMs on devices that were previously considered incapable due to resource constraints.

**Weaknesses:**

I think that although this paper is a bold attempt at the edge of LLM serving, most of the solutions provided are based on the application of past works. Firstly, whether it is tensor parallelism, sliding window, or the star-based algorithm, they are all proposed in existing and relatively mature works. The author's approach to using these methods to optimize edge LLM serving is similar to theirs, hence the paper's contribution lacks a certain level of innovation.

Furthermore, I think there are some flaws in the author's logic when explaining the motivation and opportunities for the research. I only learned from the first sentence of the abstract that the significance of serving large models at the network edge lies in data privacy-preserving. In the first paragraph of the introduction, I learned that if edge devices must be used for LLM serving to ensure user privacy and security, then more edge devices will have to be used in a distributed collaborative serving due to the limitations of computing and storage resources. However, the title of the second paragraph is "Observations and Motivations," but the content only contains observations (which should only contain them without motivations). Therefore, I suggest that the author optimize the logic of explaining the research motivation and opportunities. The scope of privacy security issues is too large, so that the research motivation seems slightly pale. Can it be combined with some more specific downstream tasks or application scenarios?

Finally, the experimental results shown in Table 1 shows that the latency of serving large models on edge devices is much higher than that in the cloud, with TTFT reaching above the second level, and the throughput is far from comparable to that of the cloud. Does this indicate that the research motivation of the paper only considered privacy protection and neglected performance issues? Although the experimental results are already much better than Galaxy. This issue is also a huge challenge that all edge LLM serving inevitably faces.

**Questions:**

When the author discusses opportunities, I have several points of confusion:

1. Why does the author believe that the communication proportion of tensor parallelism is high, but the overall inference time is reduced due to parallel computation? Is this conclusion drawn from the results in Figure 1-(b)? Has the author considered the synchronization issues among multiple edge devices in tensor parallelism?

2. Why is the total time of tensor parallelism less than 100% in Figure 1-(b)? Does the author want to use the pipeline parallelism as a baseline to illustrate the superiority of tensor parallelism among edge devices?

3. In Figure 1-(c), why is the memory footprint of each device in the TPI-LLM framework proposed in this paper the same? If tensor parallelism is used, it should decrease as the number of devices increases.

---

### Official Review · Reviewer_sPoF · 2024-11-05

**Soundness:** 1
**Presentation:** 1
**Contribution:** 1
**Rating:** 1
**Confidence:** 5

**Summary:**

The paper introduces a technique to run 70B LLM on CPU based (edge) devices. The system uses a tensor parallel framework to distribute attention heads across multiple nodes. The authors then perform an All-Reduce latency analysis, claiming that latency, not bandwidth, is the main bottleneck for all-reduce. The authors then describe a sliding memory scheduler, followed by experiments where they show the performance of their system.

**Strengths:**

The paper aims to solve a very important problem, how to run very large models with billions of parameters on CPUs or machines with no CUDA.

**Weaknesses:**

Thank you for submitting your work to ICLR. I believe the current version of the paper has many shortcomings which I will describe here in detail:

1. The paper needs thorough proof-reading. Some examples:
- by adaptively partitioning model--->a model or models
- with token latency increases-->increasing
- Write in Active voice to avoid odd sentence structures such as :
- "Constrained by the high link latency, a star-based allreduce algorithm is implemented"
- "We design a TPI-LLM"--->We design TPI-LLM
- "power collaborate"--->to collaborate
- "which dominate inference time"---> "what dominate"


2. I did not really understand the example on p.5 "For example, the path from device h2 to h1 via h8
follows the route h2 → r2 → r9 → r8 → h8 → r8 → r9 → r1 → h1, resulting in a total link latency of 16τ , where τ is the per-hop link latency"


3. The Sliding Window Memory Scheduling is very similar to PipeSwitch (https://www.usenix.org/conference/osdi20/presentation/bai). The only main difference being that you are swapping in/out from Disk to/from device memory. This is in many ways also similar to memory page to disk swapping.

4. Starting with the results for the swapping. Having an OOM is probably better than having 26.1s/token. For a 100 tokens output, you need to wait for roughly 30 minutes. This is an output of about 75 words as per OpenAI (https://help.openai.com/en/articles/4936856-what-are-tokens-and-how-to-count-them), i.e., one paragraph. Why would a user want to tolerate this? If the user chooses to use the fastest model (the Llama2-3B), they will wait for a bit less, about 3.3 minutes. I am not sure if there is a use-case for such a slow running LLM.

5. Moving to the networking results in 4.2, I think the authors are drawing the wrong conclusions. The computations are just super slow in this case that the network is not really a bottleneck in the computations. I think a better experiment would be to try the system with proper edge GPU/TPUs, e.g., Google's Corel, NVidia's Orin, or NVidia's Nano. Right now, I believe that what we are seeing is the result of just a very slow computation. You can already see that in a real world scenario, things are much worse in the real case study where the latency in Table-3 is multiple times compared to Table-1.

**Questions:**

1. What is the use-case for this work if it will take minutes to hours to generate an output?
2. How do these results change if you have a faster edge device compared to a CPU?

---

### Note · Authors · 2024-11-19

**Comment:**

We don't have the specialized edge devices with CUDA/GPUs required by the reviewer, but only general mobile devices that poor/normal users might have at home, so it's impossible for us to provide the additional experiments and data.

**Withdrawal Confirmation:**

I have read and agree with the venue's withdrawal policy on behalf of myself and my co-authors.